# Phase angle in bioelectrical impedance analysis for assessing congestion in acute heart failure

**Sangho Sohn**[1], **Jinsung Jeon**[1], **Ji Eun Lee**[2], **Soo Hyung Park**[2], **Dong Oh Kang**[2], **Eun Jin Park**[2], **Dae-In Lee**[2], **Jah Yeon Choi**[2], **Seung Young Roh**[2], **Jin Oh Na**[2], **Cheol Ung Choi**[2], **Jin Won Kim**[2], **Seung Woon Rha**[2], **Chang Gyu Park**[2], **Sunki Lee**[2]\*, **Eung Ju Kim**[2]\*

1 Department of Internal Medicine, Korea University Guro Hospital, Seoul, Republic of Korea,
2 Cardiovascular Center, Division of Cardiology, Korea University Guro Hospital, Seoul, Republic of Korea

\* galiard@naver.com (SL); withnoel@empas.com (EJK)

## Abstract

### Background

The phase angle (PhA) in bioelectrical impedance analysis (BIA) reflects the cell membrane integrity or body fluid equilibrium. We examined how the PhA aligns with previously known markers of acute heart failure (HF) and assessed its value as a screening tool.

### Methods

PhA was measured in 50 patients with HF and 20 non-HF controls along with the edema index (EI), another BIA parameter suggestive of edema. Chest computed tomography-measured lung fluid content (LFC) was used to assess pulmonary congestion. A correlation analysis was conducted to evaluate the relationships between PhA and EI, NT-proBNP, and LFC. Receiver operating characteristic (ROC) curve analysis was used to determine the cut-off values for PhA and EI for classifying patients with HF. The area under the curve (AUC) was compared using the DeLong test to evaluate the performance of PhA and EI compared to that of LFC in correctly classifying HF.

### Results

The PhA levels were significantly lower in the HF group. Whole-body PhA was 4.49° in the HF group and 5.68° in the control group. Moderate and significant correlation was observed between PhA measured at 50-kHz and both NT-proBNP (-0.56 to -0.27, all p-values<0.05) and LFC (-0.52 to -0.41, all p-values <0.05). The AUC for whole-body PhA was 0.827 (confidence interval [CI] 0.724–0.931, p<0.01) and was 0.883 (CI 0.806–0.961, p<0.01) for EI, and the optimal cutoffs were estimated as 5° (sensitivity 0.84, specificity 0.80) and 0.394 (sensitivity 0.78, specificity 0.95), respectively. When both PhA and EI were included in the model, the AUC increased to 0.905, and this was comparable to that of LFC (AUC = 0.913, p = 0.857).

**Data Availability Statement:** All relevant data are within the manuscript and its Supporting Information file (S1 File).

**Funding:** This study was financially supported by the World Class 300 Project of the Ministry of Trade, Industry and Energy (MOTIE) and the Ministry of Small and Medium-sized Enterprises and Startups (MSS) in the form of a grant (R&D, S2382763) received by EK. This study was also financially supported by Korea University in the form of a grant (K2205101) received by EK. The funders had no role in study design, data collection and analysis, decision to publish, or preparation of the manuscript.

**Competing interests:** The authors have declared that no competing interests exist.

## Conclusions

PhA exhibited a correlation with known markers of HF and demonstrated its potential as a non-invasive screening tool for the early detection of HF exacerbation. The combined use of PhA and EI can provide a robust alternative for routine self-monitoring in patients with HF, thereby enhancing early intervention.

## Introduction

Heart failure (HF) is the leading cause of initial hospitalization and rehospitalization after discharge [1]. Each episode of exacerbation accelerates functional cardiac damage and increases the risk of recurrent events, underscoring the significance of self-care, ranging from lifestyle modifications to self-monitoring [2, 3]. However, the recommended self-monitoring methods such as tracking weight gain or edema are often inaccurate and may only become evident after the clinical condition has progressed [3, 4]. Additionally, laboratory measures that are highly correlated with HF such as serum NT-proBNP levels and chest computed tomography (CT) scans that most accurately measure lung congestion require institutional facilities and professional assistance [5, 6]. Consequently, a routine self-monitoring tool that is practical in the preclinical stage remains an important requirement.

Bioelectrical impedance analysis (BIA) is based on the principle that resistance (R) and reactance ($X_C$) vary according to the properties of different media such as muscles, fat, and water when an electrical current passes through the body [7, 8]. This method allows for the comprehensive analysis of body composition, including minerals, fat-free mass, and total body water (TBW), using parameters directly measured or estimated through regression techniques and exhibits high agreement with reference methods such as dual X-ray absorptiometry [8]. Recent studies have demonstrated that the edema index (EI), an estimated BIA parameter indicating the ratio of extracellular water (ECW) to TBW, can effectively represent the edema status, including pulmonary congestion [9, 10]. Despite significant advancements in the accurate measurement and estimation of BIA, some studies suggest that directly measured parameters may be more reliable when evaluating patients who are not in normal conditions compared to estimated parameters that are based on regression models assuming normal physiological conditions [8].

The phase angle (PhA), a directly measured BIA parameter, was calculated as the arctangent of $X_C$ over R [10, 11]. $X_C$ and R are known to reflect cellular integrity and body fluid status, respectively, and studies agree that a lower PhA represents decreased intracellular water (ICW) [10, 12]. Previous studies have demonstrated that lower PhA values are associated with unfavorable morbidity and mortality across various conditions, including general health status such as frailty [13] and dysmobility syndrome [14], as well as specific diseases such as cancer [15] and diabetes [16]. In patients with HF, a low PhA is associated with a higher mortality risk and a more edematous condition [17, 18]. However, to the best of our knowledge, there is limited evidence regarding whether PhA correlates with established measures that can evaluate impending HF exacerbation such as NT-proBNP or chest CT-measured lung fluid content (LFC).

BIA is distinguished by its ease of measurement and noninvasive nature, allowing for frequent assessments as needed. If the PhA demonstrates a correlation with established HF measures, it could potentially serve as a valuable tool for routine self-monitoring to assess fluid overload and congestion during the preclinical stage.

Therefore, the purpose of this study was to evaluate the correlation between PhA measured by BIA and previously known measures of HF and to assess the additive value of PhA alongside EI as an indicator for detecting acute HF. It was hypothesized that the phase angle is significantly correlated with NT-proBNP and LFC and may serve as a useful parameter for assessing congestion in patients with acute heart failure.

## Materials and methods

### Study population

This cross-sectional observational study was conducted at Korea University Guro Hospital, Seoul, Korea, between September 2020 and January 2022. Patients admitted to or attending a cardiovascular center outpatient clinic were eligible for study recruitment. The HF group consisted of patients with HF as the primary diagnosis, exhibiting classic symptoms such as generalized edema or shortness of breath, signs including lung congestion, and a serum NT-proBNP concentration of 800 pg/mL or higher within 3 weeks. The non-HF control group included patients with no known history or current HF status. Pediatric patients aged < 20 years, pregnant women, and those with any type of cardiac implant, limb defects, or ambulatory difficulties were excluded.

### Clinical data

All data were collected at enrollment. For admitted patients, data collection was performed within three days of admission to reflect the acute status, reduce the influence of subsequent treatments, and minimize time discrepancies between tests. Upon admission, blood samples were collected to measure serum creatinine and NT-proBNP levels. Anthropometric data, including height and weight, were used to calculate body mass index (BMI). Pulmonary function tests (PFTs) were conducted using spirometry to account for differences in lung volume with respect to the LFC. Medical records were reviewed to screen for comorbidities other than cardiovascular diseases.

### Bioelectrical impedance analysis

Two BIA parameters, PhA and EI, were measured at three different frequencies (5, 50, and 250 kHz) using an InBody S10 (InBody Co., Ltd., Seoul, Korea) that is a direct segmental multi-frequency BIA (DSM-BIA) device. Compared to conventional BIA techniques such as the wrist-ankle method, DSM-BIA allows for direct measurements of the whole body as well as individual segments (arms, legs, and trunk) through multiple electrodes. The validity and test-retest reproducibility of this device, along with other DSM-BIA devices, have been demonstrated in previous studies [19–21].

Bioelectrical impedance analysis (BIA) was conducted within 3 days of admission to assess the edematous status of patients with acute HF. To reduce measurement bias, all tests were performed in a dedicated testing room by a single trained technician. Standardized pre-test precautions were implemented starting the day prior to the procedure. Patients were instructed to abstain from exercise, bathing, or activities leading to excessive sweating for 6–12 hours before the test and to avoid alcohol or caffeine for 24 hours prior. In the morning of the test, patients were asked to fast, empty their bladder, and remove all metallic or conductive items to prevent interference. BIA was performed with patients lying in the supine position on an insulated bed for 15 minutes to minimize measurement variability caused by fluid shifts. Eight electrodes were attached to the thumbs, third finger, and ankles on both sides. The patients were instructed to keep their fingers, arms, and limbs abducted to avoid electrical

current interference and to remain still during the analysis. PhA was measured segmentally for five distinct body parts (all limbs and the trunk) at all frequencies and for the whole body at 50 kHz (the most commonly used frequency).

### LFC

Chest CT scans were obtained to measure LFC using a Siemens workstation with the Syngo.CT Pulmo 3D package (Siemens Healthcare GmbH, Erlangen, Germany). The mean lung density (MLD) was determined from the CT images based on the lung attenuation distribution within the segmented sections. Pixels between 0 and 1,000 Hounsfield Units were selected. LFC was calculated as LFC (%) = (MLD + 100) ÷ 10, adjusted for lung volume data from PFT [6].

### Statistical analysis

Baseline sociodemographic and clinical characteristics are summarized as means and standard deviations, except for NT-proBNP that is presented as medians for skewness. The average PhAs from distinct segments measured at all frequencies as well as the whole-body PhA at 50 kHz were compared to determine the overall difference between the HF and non-HF controls using t-tests. To evaluate the correlation between PhA and established indicators of HF aggravation, including EI, NT-proBNP, and LFC, partial correlation coefficients were calculated. NT-proBNP was log-transformed due to its right-skewed distribution, and age, sex, BMI, and serum creatinine level were used as adjustment variables. The area under the curve (AUC) from the receiver operating characteristic (ROC) analysis was calculated to compare the discriminatory values of the parameters. First, unadjusted crude models using PhA or EI as single parameters and a multivariate model incorporating both PhA and EI were compared to an LFC single-parameter model. Next, the adjusted models, including the control variables, were compared for sensitivity analysis. The DeLong test was used for the AUC comparisons. The estimated cutoffs for PhA and EI were calculated using the corresponding validity measures. All statistical analyses were performed using SAS (version 9.4; SAS Institute Inc., Cary, NC, USA) and RStudio (RStudio Inc., Boston, MA, USA).

### Ethics

All participants were informed of the purpose and process of the study and provided written informed consent. The study design was reviewed and approved by the Institutional Review Board of Korea University Guro Hospital (2019GR0407).

## Results

Seventy patients with complete clinical and BIA data were recruited during the study period (Table 1). Fifty patients (56% men) were classified into the HF group, and 20 (60% men) were controls. In the HF group, 15 patients (30%) exhibited a preserved ejection fraction (EF), 15 (30%) possessed mid-range/mildly reduced EF, and 20 (40%) exhibited reduced EF (S1 Table). The mean ages were 70 and 57 years in the HF and control groups, respectively. The most common comorbidity in the HF group was hypertension (78%), and this was followed by diabetes (52%) and atrial fibrillation (44%). Among the non-HF controls, 75% exhibited coronary artery disease, and 70% experienced hypertension. The median NT-proBNP level in the HF group was 2962.5 pg/mL, and this was right-skewed compared to 60.2 pg/mL in the controls.

Segmental PhA was consistently lower in the HF group across all frequencies and measurement sites, except for the trunk (measured at 5 kHz) and the left arm and trunk (measured at 250 kHz) (Table 2). The whole-body PhA was 4.49˚ in the HF group and 5.68˚ in the control

**Table 1. Sociodemographic and clinical characteristics of patients with heart failure and non-heart failure controls.**

|  | Total | Patients with HF | Non-HF controls |
|---|---|---|---|
| **N (%)** | 70 (100) | 50 (71.4) | 20 (28.6) |
| **Age (years, mean [SD])** | 66.5 (14.5) | 70.3 (14.0) | 57.0 (11.0) |
| **Sex (male [%])** | 40 (57.1) | 28 (56.0) | 12 (60.0) |
| **Height (cm, mean [SD])** | 162.4 (8.9) | 161.5 (9.4) | 164.5 (7.1) |
| **Weight (kg, mean [SD])** | 65 (11.4) | 64.4 (12.4) | 66.5 (8.6) |
| **Alcohol (N [%])** | 19 (27.1) | 8 (16.0) | 11 (55.0) |
| **Smoking (N [%])** |  |  |  |
| Non-smoker | 46 (65.7) | 37 (74.0) | 9 (45.0) |
| Ex-smoker | 9 (12.9) | 4 (8.0) | 5 (25.0) |
| Current-smoker | 15 (21.4) | 9 (18.0) | 6 (30.0) |
| **Clinical characteristics** |  |  |  |
| Comorbidities (N [%]) |  |  |  |
| **Hypertension** | 53 (75.7) | 39 (78.0) | 14 (70.0) |
| **Diabetes** | 32 (45.7) | 26 (52.0) | 6 (30.0) |
| **Dyslipidemia** | 22 (31.4) | 13 (26.0) | 9 (45.0) |
| **Coronary artery disease** | 35 (50.0) | 20 (40.0) | 15 (75.0) |
| **Atrial fibrillation** | 22 (31.4) | 22 (44.0) | 0 (0) |
| **COPD** | 4 (5.7) | 4 (8.0) | 0 (0) |
| **Chronic kidney disease** | 18 (25.7) | 18 (36.0) | 0 (0) |
| **Laboratory findings** |  |  |  |
| NT-proBNP (pg/mL, median [SD]) | 1811 (5701.5) | 2962.5 (6242.1) | 60.2 (46.8) |
| Creatinine (mg/dL, mean [SD]) | 1.2 (0.9) | 1.3 (1.0) | 0.9 (0.2) |

**COPD**: Chronic obstructive pulmonary disease; **HF**: Heart failure; **SD**: Standard deviation

group, indicating a significant difference (p<0.001). At 50 kHz, all measured segments exhibited significantly lower PhA values in the HF group. The trunk exhibited significant differences only at 50 kHz, although the PhAs at 5 kHz and 250 kHz also tended to be lower in the HF group (2.36˚ and 3.40˚, respectively) than they were in the control group (2.64˚ and 3.68˚, respectively). The PhA measured in the legs was consistently lower in the HF group at all frequencies. Additionally, the differences in whole-body and lower-limb PhAs measured at 50 kHz were robust across all HF phenotypes (S2 Table).

The correlations between PhA and other measures of acute HF ranged from moderate to strong (Fig 1). The correlation with EI, another BIA parameter, was particularly strong, and

**Table 2. Comparison of segmental phase angles in different frequencies measured by bioelectrical impedance analysis in patients with heart failure and in non-heart failure controls.**

| Segmental phase angle (˚) | 5 kHz | | | 50 kHz | | | 250 kHz | | |
|---|---|---|---|---|---|---|---|---|---|
|  | HF | Control | p-value* | HF | Control | p-value* | HF | Control | p-value* |
| **Whole body** |  |  |  | 4.49 | 5.68 | <0.001 |  |  |  |
| **Right arm** | 1.90 | 2.32 | 0.003 | 4.61 | 5.43 | 0.001 | 5.64 | 6.10 | 0.014 |
| **Left arm** | 1.89 | 2.19 | 0.040 | 4.50 | 5.08 | 0.011 | 5.62 | 5.61 | 0.917 |
| **Trunk** | 2.36 | 2.64 | 0.088 | 4.33 | 5.33 | 0.001 | 3.40 | 3.68 | 0.611 |
| **Right leg** | 1.88 | 2.66 | <0.001 | 4.28 | 6.08 | <0.001 | 3.47 | 4.18 | 0.002 |
| **Left leg** | 1.88 | 2.66 | <0.001 | 4.18 | 6.12 | <0.001 | 3.28 | 4.44 | <0.001 |

* Compared using independent *t*-test; **HF**: Heart failure; **kHz**: kilohertz.

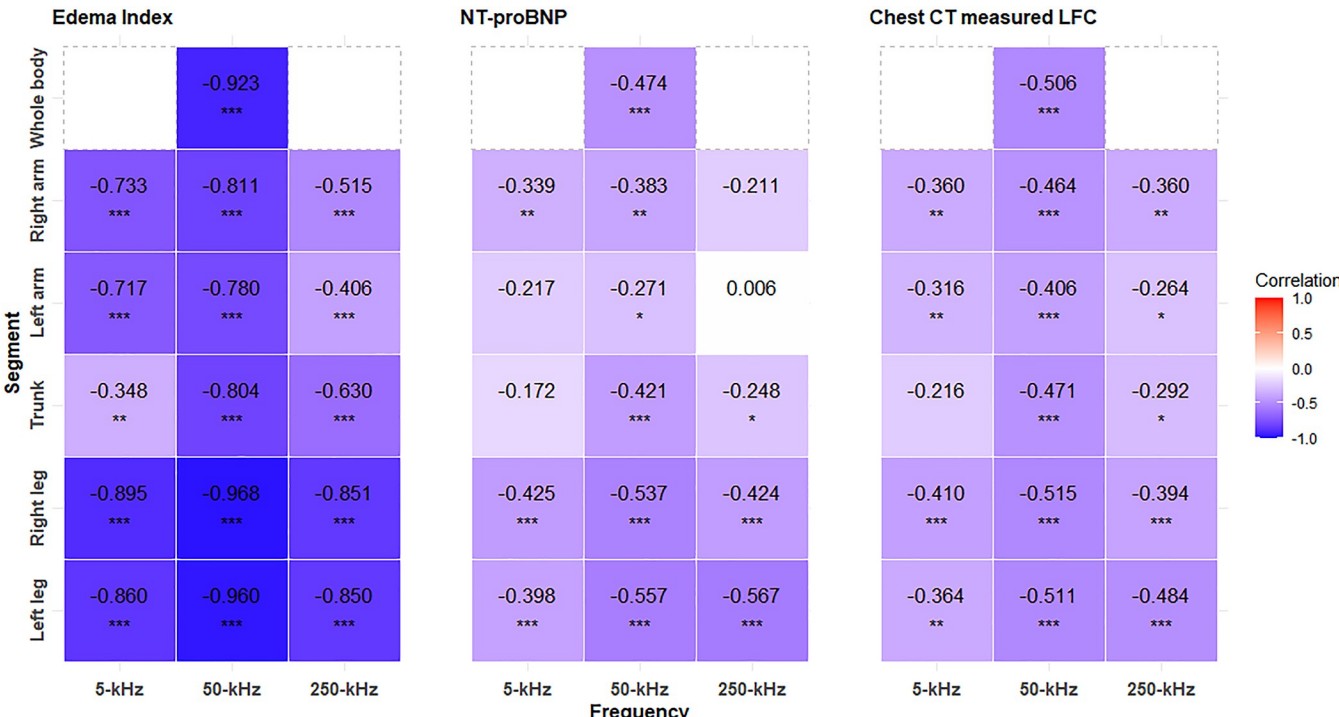

**Fig 1. Correlation between the phase angle and bioelectrical impedance analysis-measured edema index, serum NT-proBNP, and chest CT-measured lung fluid content (LFC).** Numbers indicate coefficient values, and marks indicate p-values for partial correlation controlled for age, sex, body mass index, and serum creatinine. (*: <0.05, **: <0.01, and ***: <0.001).

comparisons between NT-proBNP and the gold standard, LFC, also demonstrated moderate results. The correlation between whole-body PhA and EI was the highest (-0.923), and comparisons to NT-proBNP and LFC remained significant (-0.474 and -0.506, respectively). In the arms and trunk, the correlations between NT-proBNP and LFC did not exhibit statistical significance at 5 and 250 kHz, similar to the simple comparisons observed in Table 2. Nevertheless, the overall trend of the negative correlations persisted, with comparisons at 50 kHz consistently demonstrating significant differences. The strongest correlations were observed in both legs across all frequencies and compared parameters, maintaining statistical significance. The coefficients and p-values are listed in S3 Table.

PhA independently exhibited a discriminatory value for HF with an AUC of 0.827 (confidence interval [CI]: 0.724–0.931), and this was similar to that of EI (AUC = 0.883, CI: 0.806–0.961) (Fig 2). Interestingly, when both PhA and EI were included in the model, the performance increased to AUC = 0.905 (CI: 0.835–0.975), and this was comparable to that of the LFC model (AUC = 0.913; CI: 0.849–0.977), with no significant difference (p = 0.857).

The AUC analysis remained significant after adjusting for variables (S1 Fig). As expected, the AUC increased for all measures, as more explanatory variables were added. However, the difference between the adjusted PhA and EI models (AUC = 0.923) and the adjusted LFC model (AUC = 0.968, p = 0.210) was not significant.

The optimal cut-off for PhA for distinguishing HF aggravation from other conditions was estimated to be 5.0˚, with 84% sensitivity and 80% specificity (Table 3). The specificity of LFC was particularly high at 99%, with a cut-off value of 18.50.

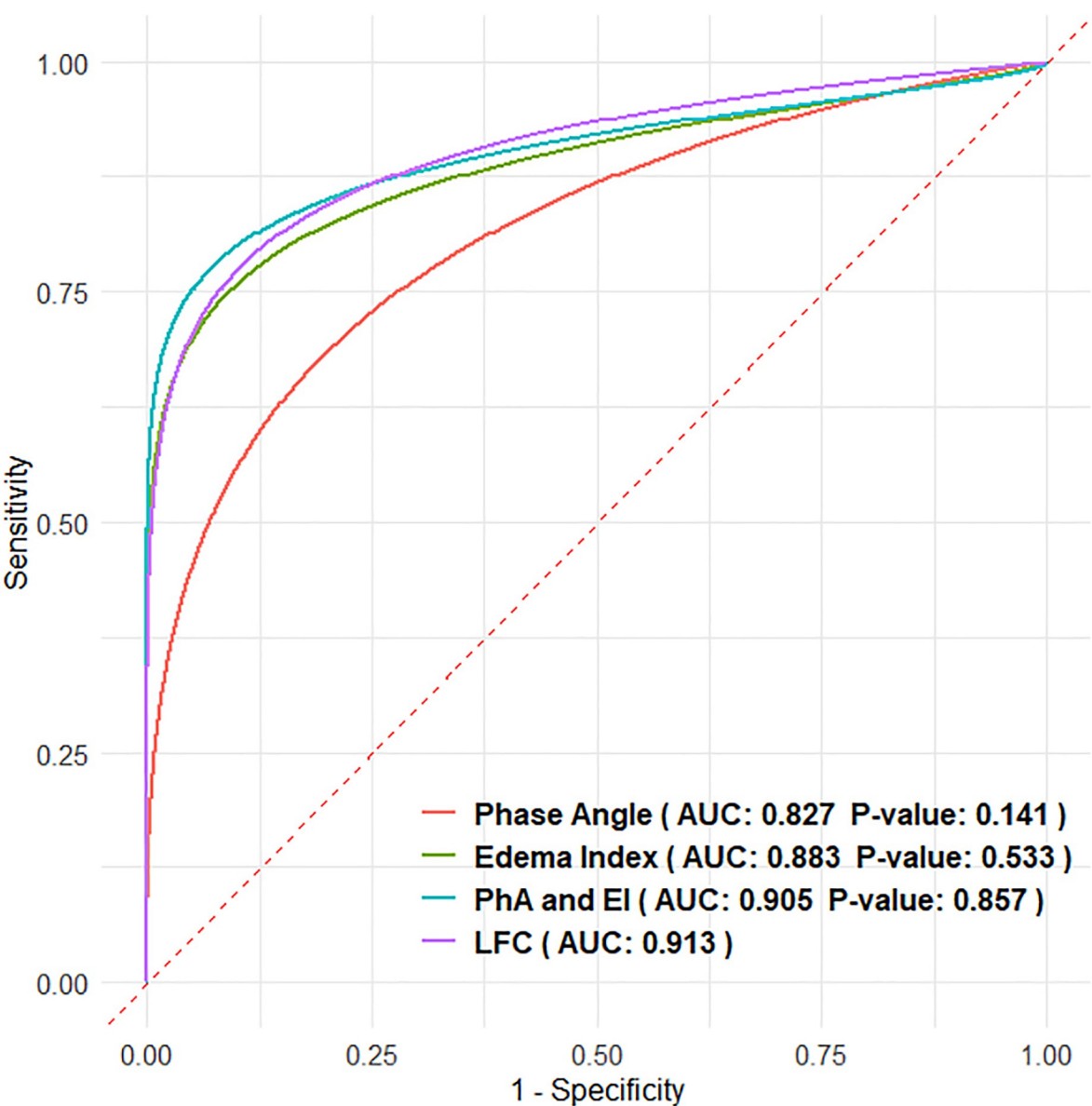

**Fig 2. Area under the curves of phase angle (PhA), edema index (EI), combined PhA and EI, and chest CT-measured lung fluid content (LFC).** P-values were calculated using the DeLong test for comparison of each measure versus the LFC as reference.

**Table 3. Estimated cutoff values and validity indices for phase angle, edema index, and chest CT-measured lung fluid content regarding heart failure.**

|  | AUC (CI) | p-value | Cutoff | Sensitivity | Specificity |
|---|---|---|---|---|---|
| Phase angle | 0.827 (0.724–0.931) | <0.001 | 5.00 | 0.84 | 0.80 |
| Edema index | 0.884 (0.806–0.961) | <0.001 | 0.40 | 0.78 | 0.95 |
| Lung fluid content | 0.913 (0.849–0.977) | <0.001 | 18.50 | 0.74 | 0.99 |

**AUC**: Area under the curve; **CI**: Confidence interval

## Discussion

This study examined the relationship between PhA, a parameter easily measured using BIA, and previously known markers of HF aggravation to evaluate its feasibility as an early indicator. The results indicate that PhA was consistently lower in patients with HF, suggesting a correlation with congestion from acute HF. Additionally, PhA exhibited moderate to strong correlations with EI, serum NT-proBNP levels, and chest CT-measured LFC. The discriminatory value of PhA was estimated to be approximately 83%, and when PhA and EI were considered together, their combined performance was comparable to that of LFC.

While EI is understood as the derived estimation of the distribution between ECW and TBW, PhA differs in that it directly measures the integrity of the cellular membrane that causes such a distribution [7]. When the cellular membrane integrity is compromised, the body fluid equilibrium shifts toward the ECW from the ICW. Particularly, in patients with HF this condition appears as an edematous symptom, and BIA measured around this point results in a lower PhA [10]. Previous studies have reported that the PhA was approximately 4.7˚ in patients with acute HF and between 4.2˚ and 4.5˚ in patients with HF and edema [18]. A previous study of patients with New York Heart Association class III or IV HF reported PhAs ranging from 4.2˚ to 4.9˚ in systolic HF and 4.2˚ to 4.8˚ in diastolic HF [22]. This study demonstrated similarly low PhA values of 4.5˚ in patients with HF (Table 2), and it is consistent with the findings of previous research. Unlike previous studies that focused solely on patients with HF, this study included a comparison group with other cardiac diseases, and PhA still exhibited a clear distinction between patients with and without HF. It suggests the potential utility of PhA as an assessment tool for congestion that indicates HF aggravation, among other acute conditions.

Apparent differences in the segmental PhA were observed in the lower limbs that were consistent across all frequencies (Table 2). In a study comparing EI, another BIA-estimated parameter, in patients with HF, the largest difference was observed in the lower limbs [9]. The authors explained that fluid shift to the lower limbs may have resulted in such an observation based on the patients being ambulatory, and this is an interpretation that can also be applied to current findings. Additionally, more than 55% of skeletal muscle mass is located in the lower limbs, and BIA is known to reflect this distribution [21, 23]. As both PhA and EI are influenced by the distribution of water within or outside the cells, it is reasonable that the results were particularly robust in the lower limbs that possess more cellular components.

Correlations between PhA and established measures of acute HF, including NT-proBNP and LFC, were moderate, regardless of the measured body segments (Fig 1). Although an overall negative trend persisted, a few segments such as the trunk and arm did not reach statistical significance at 5 kHz and 250 kHz. BIA parameters were measured based on the theory that at zero frequency, the current cannot pass through the cell and thus flows only through the ECW, whereas at very high frequencies, the current penetrates all body components [7, 12, 24]. Therefore, measurements below 5 kHz or above 200 kHz are known to be less accurate [11], whereas midrange frequencies provide the highest measurement accuracy, particularly in diseased subjects [25]. Additionally, although the trunk constitutes approximately 60% of the body mass, it accounts for only 25% of the total body R, thus making it prone to measurement errors [26]. This explains why previous studies commonly used whole-body PhA measurements at 50 kHz [18, 27, 28].

This study demonstrated that a PhA cut-off estimation of 5˚ provided the maximum AUC (Table 3), and it was slightly higher than the average PhA of 4.18˚–4.61˚ observed in the patient group. Interestingly, the sensitivity of 84% at this cutoff value was higher than the sensitivity values for EI and LFC at their respective cutoffs. Studies have demonstrated varying

reference ranges for PhA, even in healthy populations [29], due to differences in age and sex, making it more challenging to establish a definitive PhA cut-off for disease conditions [18]. However, using PhA to assess congestion is expected to facilitate frequent monitoring, enabling early detection of changes in fluid status and timely interventions such as diuretic adjustments that may help prevent readmissions and improve patient outcomes [4]. In this context, a broader risk range (*i.e.*, a higher PhA), although less precise, may offer better preventive benefits than a narrower value that might overlook patients at borderline risk.

## Limitations

This study possesses some limitations. First, only one of the many devices available to measure BIA parameters was used. Therefore, the consistency of the results across different devices could not be verified. Some studies have reported discrepancies when using different BIA devices in the same individual [30, 31]. Although the results from different devices for the same individuals were not in exact agreement, the studies indicated strong agreement between devices on group-level results such as mean or distribution [31]. The findings commonly concluded that BIA parameters are still reliable for longitudinal use in tracking changes in individuals, as long as devices are not used interchangeably during follow-up. Therefore, as the focus of current study was on the feasibility of using PhA for home monitoring rather than for confirmatory diagnosis, inter-device differences should not be a major concern.

Second, this study was conducted at a single center with a small number of patients. Therefore, it was difficult to perform further subgroup analyses such as those assessing possible differences between the phenotypes of HF or the underlying conditions. Additionally, as this study serves as a preliminary exploration of the potential utility of relatively novel markers such as the phase angle where supporting evidence remains limited, determining an a priori sample size was not feasible, and this is a common challenge in early-stage research [6, 9, 32]. To mitigate this limitation, the study enrolled as many participants as possible within the study period. However, expanding the scope of the study presented additional challenges due to the need for control participants to undergo chest CT scans that exposes them to unnecessary radiation, despite the low actual risk involved. Despite these difficulties, the major findings, including the significant differences in PhA between the groups and low values in patients with HF, are consistent with those of previous studies [18, 22]. Considering the conceptualization stage of current research, the implications of the results are assumed to be reasonable.

Third, the cross-sectional design of the study limited ability to evaluate whether PhA or other markers such as NT-proBNP, measured at a single time point or tracked over time could offer insights into the clinical outcomes and prognosis of patients with HF. Further studies utilizing time-series data are required to understand the clinical implications of PhAs, both at the preclinical stage and over the course of long-term management.

## Conclusion

The PhA measured using BIA is a valuable parameter that correlates well with established markers of HF. Its noninvasive nature and reliable accuracy underscore its potential as an additional tool for assessing fluid overload and congestion in patients with HF, making it suitable for daily monitoring by non-professionals. Furthermore, the combined use of PhA and other BIA parameters such as EI could offer a robust alternative to invasive or complex measures at the prehospital stage.

## Supporting information

**S1 Table. Phenotypes of heart failure based on ejection fraction in the patient group.** EF: Ejection fraction; HF: Heart failure; HFmrEF: Heart failure with mid-range/mildly reduced ejection fraction; HFpEF: Heart failure with preserved ejection fraction; HFrEF: Heart failure with reduced ejection fraction.
(DOCX)

**S2 Table. Comparison of segmental phase angles by different phenotypes of heart failure.** * Measured at 50 kilo-Hertz; ** Compared using independent *t*-test; HFmrEF: Heart failure with mid-range/mildly reduced ejection fraction (40%<EF<50%); HFpEF: Heart failure with preserved ejection fraction (EF≥50%); HFrEF: Heart failure with reduced ejection fraction (EF≤40%).
(DOCX)

**S3 Table. Correlation between bioelectrical impedance analysis-measured phase angle and edema index and other established markers of heart failure including serum NT-proBNP and chest CT-measured lung fluid content.** * Partial correlation coefficient adjusted for age, sex, body mass index, and serum creatinine; BNP: log-transformed serum NT-proBNP; EI: Edema index; kHz: kilohertz; LFC: chest CT-measured lung fluid content.
(DOCX)

**S1 Fig. Area under the curves of phase angle (PhA), edema index (EI), combined PhA and EI, and chest CT-measured lung fluid content (LFC) after adjusting for age, sex, body mass index, and serum creatinine level.** P-values were calculated using the DeLong test to compare each measure versus LFC as a reference.
(TIF)

**S1 File.**
(CSV)

## Author Contributions

**Conceptualization:** Sangho Sohn, Sunki Lee, Eung Ju Kim.

**Data curation:** Sangho Sohn, Ji Eun Lee, Soo Hyung Park, Dong Oh Kang, Eun Jin Park, Dae-In Lee, Jah Yeon Choi, Seung Young Roh, Jin Oh Na, Cheol Ung Choi, Jin Won Kim, Seung Woon Rha, Chang Gyu Park, Sunki Lee, Eung Ju Kim.

**Formal analysis:** Sangho Sohn, Jinsung Jeon, Sunki Lee, Eung Ju Kim.

**Funding acquisition:** Sunki Lee, Eung Ju Kim.

**Investigation:** Sangho Sohn, Sunki Lee, Eung Ju Kim.

**Methodology:** Sangho Sohn, Jinsung Jeon, Sunki Lee, Eung Ju Kim.

**Project administration:** Sangho Sohn, Sunki Lee, Eung Ju Kim.

**Resources:** Sangho Sohn, Sunki Lee, Eung Ju Kim.

**Software:** Sangho Sohn, Sunki Lee, Eung Ju Kim.

**Supervision:** Sunki Lee, Eung Ju Kim.

**Validation:** Sangho Sohn, Jinsung Jeon, Sunki Lee, Eung Ju Kim.

**Visualization:** Sangho Sohn, Sunki Lee, Eung Ju Kim.

**Writing – original draft:** Sangho Sohn.

**Writing – review & editing:** Sangho Sohn, Jinsung Jeon, Ji Eun Lee, Soo Hyung Park, Dong Oh Kang, Eun Jin Park, Dae-In Lee, Jah Yeon Choi, Seung Young Roh, Jin Oh Na, Cheol Ung Choi, Jin Won Kim, Seung Woon Rha, Chang Gyu Park, Sunki Lee, Eung Ju Kim.

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
