## [Decision Letter · Decision Letter 0]

7 Oct 2024

PONE-D-24-38284Phase angle in bioelectrical impedance analysis: a Promising indicator for acute heart failure screeningPLOS ONE

Dear Dr. Lee,

Thank you for submitting your manuscript to PLOS ONE. After careful consideration, we feel that it has merit but does not fully meet PLOS ONE’s publication criteria as it currently stands. Therefore, we invite you to submit a revised version of the manuscript that addresses the points raised during the review process.

**ACADEMIC EDITOR:**Dear authors,

The manuscript ONE-D-24-38284, "Phase angle in bioelectrical impedance analysis: a Promising indicator for acute heart failure screening", is very interesting and relevant to state of art while also a hot topic.

However, the current form can't be fully replicated which is a major rule for science reprodutibility.

In addition to the comments made by reviewers, please see the following:

-abstract should include statistical results;

-keywords should not contain the same words already present in the title;

-a hypothesis should be added in the end of introduction (it should be supported with references);

-a sample size calculation or statistical power should be added in methods;

-just like reviewers 1 mentioned, authors are welcome to use STROBE observational studies reporting guidelines https://www.strobe-statement.org/checklists/ to verify if there is an additional aspect to include or modify. For instance, described the setting and location of subject recruitment and sample size calculations;

-information about the procedures adopted before the body compositon analysis should be described as well as the timming of the assessment;

-validity of the device used can be added as well;

-for better clarity, authors are welcome to introduce a subsection in discussion with limitations, practical implications and directions for future studies. A final section of conclusions would also bring clarity for the reader. This topics are not mandatory in this journal, but they can improve clarity and quality of the work.

Best regards 

We look forward to receiving your revised manuscript.

Kind regards,

Rafael Franco Soares Oliveira

Academic Editor

PLOS ONE

Journal Requirements: When submitting your revision, we need you to address these additional requirements. 1. Please ensure that your manuscript meets PLOS ONE's style requirements, including those for file naming. The PLOS ONE style templates can be found at https://journals.plos.org/plosone/s/file?id=wjVg/PLOSOne_formatting_sample_main_body.pdf and https://journals.plos.org/plosone/s/file?id=ba62/PLOSOne_formatting_sample_title_authors_affiliations.pdf 2. Thank you for stating the following financial disclosure: "This research was funded by the World Class 300 Project (R&D, S2382763) of the Ministry of Trade, Industry and Energy (MOTIE) and the Ministry of Small and Medium-sized Enterprises and Startups (MSS), and partially funded by Korea University (K2205101)"  Please state what role the funders took in the study.  If the funders had no role, please state: ""The funders had no role in study design, data collection and analysis, decision to publish, or preparation of the manuscript."" If this statement is not correct you must amend it as needed. Please include this amended Role of Funder statement in your cover letter; we will change the online submission form on your behalf. 3. Thank you for stating in your Funding Statement: "This research was funded by the World Class 300 Project (R&D, S2382763) of the Ministry of Trade, Industry and Energy (MOTIE) and the Ministry of Small and Medium-sized Enterprises and Startups (MSS), and partially funded by Korea University (K2205101)" Please provide an amended statement that declares *all* the funding or sources of support (whether external or internal to your organization) received during this study, as detailed online in our guide for authors at http://journals.plos.org/plosone/s/submit-now.  Please also include the statement “There was no additional external funding received for this study.” in your updated Funding Statement. Please include your amended Funding Statement within your cover letter. We will change the online submission form on your behalf. 4. In the online submission form, you indicated that "Data cannot be shared publicly because the informed consent from participants did not include data sharing clause. However, it may be considered upon reasonable request." All PLOS journals now require all data underlying the findings described in their manuscript to be freely available to other researchers, either 1. In a public repository, 2. Within the manuscript itself, or 3. Uploaded as supplementary information.This policy applies to all data except where public deposition would breach compliance with the protocol approved by your research ethics board. If your data cannot be made publicly available for ethical or legal reasons (e.g., public availability would compromise patient privacy), please explain your reasons on resubmission and your exemption request will be escalated for approval. 5. PLOS requires an ORCID iD for the corresponding author in Editorial Manager on papers submitted after December 6th, 2016. Please ensure that you have an ORCID iD and that it is validated in Editorial Manager. To do this, go to ‘Update my Information’ (in the upper left-hand corner of the main menu), and click on the Fetch/Validate link next to the ORCID field. This will take you to the ORCID site and allow you to create a new iD or authenticate a pre-existing iD in Editorial Manager.

Additional Editor Comments:

Dear authors,

The manuscript ONE-D-24-38284, "Phase angle in bioelectrical impedance analysis: a Promising indicator for acute heart failure screening", is very interesting and relevant to state of art while also a hot topic.

However, the current form can't be fully replicated which is a major rule for science reprodutibility.

In addition to the comments made by reviewers, please see the following:

-abstract should include statistical results;

-keywords should not contain the same words already present in the title;

-a hypothesis should be added in the end of introduction (it should be supported with references);

-a sample size calculation or statistical power should be added in methods;

-just like reviewers 1 mentioned, authors are welcome to use STROBE observational studies reporting guidelines https://www.strobe-statement.org/checklists/ to verify if there is an additional aspect to include or modify. For instance, described the setting and location of subject recruitment and sample size calculations;

-information about the procedures adopted before the body compositon analysis should be described as well as the timming of the assessment;

-validity of the device used can be added as well;

-for better clarity, authors are welcome to introduce a subsection in discussion with limitations, practical implications and directions for future studies. A final section of conclusions would also bring clarity for the reader. This topics are not mandatory in this journal, but they can improve clarity and quality of the work.

Best regards

Reviewers' comments:

Reviewer's Responses to Questions

**Comments to the Author**

1. Is the manuscript technically sound, and do the data support the conclusions?

Reviewer #1: No

Reviewer #2: Yes

2. Has the statistical analysis been performed appropriately and rigorously? 

Reviewer #1: No

Reviewer #2: Yes

3. Have the authors made all data underlying the findings in their manuscript fully available?

Reviewer #1: No

Reviewer #2: Yes

4. Is the manuscript presented in an intelligible fashion and written in standard English?

Reviewer #1: No

Reviewer #2: Yes

5. Review Comments to the Author

Reviewer #1: PONE-D-24-38284

Phase angle in bioelectrical impedance analysis: a Promising indicator for acute heart failure screening."

I suggest authors use STROBE observational studies reporting guidelines https://www.strobe-statement.org/checklists/ to verify if there is an additional aspect to include or modify. For instance, described the setting and location of subject recruitment and sample size calculations.

This study had two objectives, but the authors presented them as one. To evaluate a screening tool, it is necessary to perform a different methodology (diagnostic test and longitudinal) and study populations with follow-up to develop acute heart failure.

The equipment used to evaluate the phase angle (InBody S10 analyzer) provides segmental measurements that are calculated or modelled to obtain total body parameters, which may not be appropriate biophysically; therefore, data from this device could be unreliable or underestimated. In addition, this device's measurements at the lowest and highest frequencies can interfere with other devices, such as cardiorespiratory monitoring used in heart failure patients, giving erroneous results that the evaluator does not detect.

The validity of BIA measurements when, where, and how they were performed was not informed.

A phase angle cut off value of 5 is very high the normal range goes from 5.03 to 7 and in reported phase angle values associated with negative outcomes <4°

Because phase angle (PhA), edema index, and chest CT-measured lung fluid content are correlated, should not be evaluated as “predictors” because of multicollinearity.

The areas under the curves should be presented with confidence intervals and a comparison made using the DeLong test.

Reviewer #2: In the study, the authors found that the phase angle in bioelectrical impedance analysis was significantly lower in the HF group individuals and may be associated with HF exacerbations due to fluid retention. The findings seem to impressive and practically useful. However, I would like to make some comments.

1. It remaines unclear what phenotype(s) of HF was /were included in the study.

2. How did the serum NT-proBNP concentration change in oncervation period and did the authors evaluate the link between the phase angle and NT-proBNP.

3. The patients' cohor was geterogenic and composes of AF individuals and diabetes mellitus. Did the authors investigate whether the phase angle is able to predict HF exacerbation in individuals as with lower NT-proBNP (i.e., patients with diabetes) as well as individuals with higher than expected (AF patients)?

6. PLOS authors have the option to publish the peer review history of their article (what does this mean?). If published, this will include your full peer review and any attached files.

Reviewer #1: No

Reviewer #2: No

---

## [Author Response · Author response to Decision Letter 0]

29 Oct 2024

<Response for Reviewer #1>

1. I suggest authors use STROBE observational studies reporting guidelines https://www.strobe-statement.org/checklists/ to verify if there is an additional aspect to include or modify. For instance, described the setting and location of subject recruitment and sample size calculations.

As your comment, we have thoroughly reviewed our entire manuscript and revised it in greater detail to comply with the STROBE guideline. In response to your suggestions, we have uploaded the STROBE checklist with this revision.

We agree that performing a proper sample size calculation could have strengthened our study. However, as our study is exploratory in nature, closer to a pilot or preliminary investigation rather than confirmatory research, it was challenging to postulate reference values needed for sample size calculations. This difficulty is often seen in early-stage, conceptual studies (Amir et al., 2016; Lee et al., 2024; Leungratanamart et al., 2024). We added this issue to the study limitations in the Discussion section as follows:

(Page 15/Lines 275-281) As this study serves as a preliminary exploration of the potential utility of relatively novel markers, such as phase angle, where supporting evidence remains limited, determining an a priori sample size was not feasible, a common challenge in early-stage research [6,9,32]. To mitigate this limitation, we aimed to enroll as many participants as possible within the study period. However, expanding the study’s scope presented additional challenges due to the need for control participants to undergo chest CT scans, which expose them to unnecessary radiation, despite the low actual risk involved.

2. This study had two objectives, but the authors presented them as one. To evaluate a screening tool, it is necessary to perform a different methodology (diagnostic test and longitudinal) and study populations with follow-up to develop acute heart failure.

Thank you for your valuable comment. We have carefully considered this critical comment and agree that our original submission could indeed be misleading. Our intention in using terms such as ‘screening’ or ‘predicting’ was to highlight the potential role of phase angle in identifying patients at risk of acute HF, rather than implying its use for confirmatory diagnosis. In accordance to your feedback and to avoid any misleading to readers, we decided to change and clarify the title of the study as follows:

(Page 1/Lines 1-2) Phase angle in bioelectrical impedance analysis for assessing congestion in acute heart failure

Additionally, we have narrowed the focus of the study’s purpose to emphasize identifying the correlation between phase angle and markers of congestion, while explaining the clinical implications related to acute HF as follows:

(Page 5/Lines 87-79) Therefore, the purpose of this study was to evaluate the correlation between the PhA measured by BIA and previously known measures of HF, and to assess the additive value of PhA alongside the EI as an indicator for detecting acute HF.

Furthermore, we have made extensive revisions throughout the manuscript, particularly in the Discussion section, to remove any ambiguous terms and better clarify our intention. For example:

(Page 13/Lines 232-233) This suggests the potential utility of PhA as an assessment tool for congestion that indicates HF aggravation among other acute conditions.

(Page 14/Lines 258-261) However, using PhA to assess congestion is expected to facilitate frequent monitoring, enabling early detection of changes in fluid status and timely interventions, such as diuretic adjustments, which may help prevent readmissions and improve patient outcomes.

(Page 15/Lines 293-295) Its non-invasive nature and reliable accuracy underscore its potential as an additional tool for assessing fluid overload and congestion in HF patients, making it suitable for daily monitoring by nonprofessionals.

3. The equipment used to evaluate the phase angle (InBody S10 analyzer) provides segmental measurements that are calculated or modelled to obtain total body parameters, which may not be appropriate biophysically; therefore, data from this device could be unreliable or underestimated.

Thank you for raising this important point. We were also aware of the concerns surrounding different BIA measuring techniques, especially the use of complex regressions or modeling in some earlier BIA devices. Many of these devices assumed of a single-cylinder model, providing a single whole-body measurement, which could introduce inaccuracies.

However, the InBody S10 in our study employs a more advanced technique known as direct segmental multi-frequency bioelectrical impedance analysis (DSM-BIA). DSM-BIA allows for direct measurement of individual body segments (arms, legs, and trunk) in addition to the whole body, minimizing the potential for errors arising from generalized assumptions or modeling processes. By avoiding the error-prone calculations used in earlier techniques, DSM-BIA is known to provide more accurate and reliable data across different body segments, reducing the likelihood of misrepresenting parameters.

Furthermore, the validity and reproducibility of DSM-BIA devices, including the InBody S10, have been extensively studied. Multiple studies have demonstrated their reliability in both validation and clinical applications, with reasonable agreement across various devices from different manufacturers (Ling et al., 2011; Bosy-Westphal et al., 2017; Yang et al., 2017). Considering this context, we would like to say that the selection of certain device that has already been widely accepted is less likely to compromise the integrity of our findings.

In response to your concern, however, we have revised the Methods section to include additional details on the measurement procedure and device selection, as follows:

(Page 6/Lines 112-116) Two BIA parameters, PhA and EI, were measured at three different frequencies (5 kHz, 50 kHz, and 250 kHz) using the InBody S10 (InBody Co., Ltd., Seoul, Korea), a direct segmental multi-frequency BIA (DSM-BIA) device. Compared to conventional BIA techniques such as the wrist-ankle method, DSM-BIA allows for direct measurements of the whole body as well as individual segments (arms, legs, and trunk) through multiple electrodes. The validity and test-retest reproducibility of this device, along with other DSM-BIA devices, have been demonstrated in previous studies [19–21].

4. In addition, this device's measurements at the lowest and highest frequencies can interfere with other devices, such as cardiorespiratory monitoring used in heart failure patients, giving erroneous results that the evaluator does not detect.

We regret to mention this point more in detail. As mentioned in our original manuscript under the exclusion criteria (“patients with any type of cardiac implants, limb defects, or ambulatory difficulties were excluded”), we had already taken this issue into consideration during the study design phase to avoid any potential interference from the BIA device with other monitoring devices. Additionally, we have now provided more detailed information on the procedure in the Methods section to further clarify this aspect as follows:

(Page 6/Line 120) All metallic or conductive objects and personal items were removed to avoid interference.

5. The validity of BIA measurements when, where, and how they were performed was not informed.

We described more in detail regarding the measurement process as follows:

(Pages 6-7/Lines 117-123) To better capture the edematous status in acute HF, BIA was performed within 3 days of admission. The test was conducted in a dedicated testing room by a single trained technician to minimize measurement bias. Before the exam, patients rested in a supine position on an insulating bed for 15 min. All metallic or conductive objects and personal items were removed to avoid interference. Eight electrodes were attached to the thumbs, third fingers, and ankles on both sides. Patients were instructed to keep their fingers, arms, and limbs abducted to avoid electrical current interference and remain still during the analysis.

Additionally, in response to your valuable feedback on the BIA measurement, we have revised the Bioelectrical Impedance Analysis section in the Methods to provide more comprehensive details. Thank you for your insightful comments.

6. A phase angle cut off value of 5 is very high the normal range goes from 5.03 to 7 and in reported phase angle values associated with negative outcomes <4°

Thank you for this insightful comment. While it is challenging to establish a definitive reference value or range for phase angle, we agree that the generally accepted normal range is above 5°, which overlaps with our suggested cutoff value. However, evidence indicates that normal values can vary depending on factors such as age, gender, and underlying conditions (Mattiello et al., 2020). Likewise, the abnormal values also differ based on whether the condition is acute or chronic, even within patients with HF from 4.7° to 5.5° (Massari et al., 2016).

Although the cutoff value of 5° may appear higher, the average values in our study ranged between 4.18° and 4.61°, which fall within a reasonable range compared to similar studies, supporting the validity of our results. Since our study suggests frequent self-monitoring of PhA by patients for early detection of edematous conditions, we believe that providing a broader, albeit less precise, risk range (e.g., higher PhA) may be more advantageous in a preventive setting compared to a narrower, more exact value.

We recognize that this is a clinically important issue and have added further details to the Discussion section to address it as follows:

(Page 14/Lines 253-262) Our study demonstrated that a PhA cutoff estimation of 5º provided the maximum AUC (Table 3), which was slightly higher than the average PhA of 4.18° to 4.61° observed in the patient group. Interestingly, the sensitivity of 84% at this cutoff was higher than the sensitivity values for EI and LFC at their respective cutoffs. Studies have shown varying reference ranges for PhA, even in healthy populations [29], due to differences in age and sex, making it more challenging to establish a definitive PhA cutoff for disease conditions [18]. However, the expected implication of using PhA for assessing congestion is to enable frequent monitoring, allowing for the early detection of fluid status changes and the initiation of prompt interventions, such as adjusting diuretics, which can prevent readmissions and improve prognosis [4]. In this context, a broader risk range (i.e., higher PhA), though less precise, may offer better preventive benefits than a narrower value, which might overlook patients at borderline risk.

7. Because phase angle (PhA), edema index, and chest CT-measured lung fluid content are correlated, should not be evaluated as “predictors” because of multicollinearity.

We agree that the measures reviewed in our study, such as PhA, EI, and chest CT-measured lung fluid content, are correlated, and indeed, the purpose of our study was to examine this correlation. We also acknowledge that correlated variables should be handled carefully, as you mentioned. However, to the best of our understanding, multicollinearity is primarily a concern when estimating coefficients through regression models, which differs from the objective of our study.

In our analysis, we used PhA and EI as variables in the ROC analysis, which aimed to assess their ability to classify study subjects according to the appropriate disease status. We did not perform point estimation of these variables' effect sizes, where multicollinearity would be more relevant.

That being said, we do agree that our original use of the term ‘predictors’ may have been misleading to some readers. Therefore, we have revised our terminology, replacing ‘predictors’ with ‘parameters’ to reflect a more neutral interpretation as follows:

(Page 7/Lines 139-143) Area under the curve (AUC) from the receiver operating characteristic analysis was calculated to compare the discriminatory value of parameters. First, unadjusted crude models using PhA or EI as single parameters and a multivariable model incorporating both PhA and EI were compared to an LFC single parameter model. Next, adjusted models including control variables were compared for sensitivity analysis.

8. The areas under the curves should be presented with confidence intervals and a comparison made using the DeLong test.

Thank you for your comment. We have updated Table 3 to include confidence intervals for the AUC values, as recommended.

Additionally, the DeLong test was performed for AUC comparisons in our original analysis, but we regret that this was not detailed sufficiently. We have now corrected this in the Methods section and added further details in the footnotes of the relevant figures as follows:

(Page 7/Line 143) DeLong test was used for these AUC comparisons.

(Page 22/Lines 423-424) p-values were calculated using the DeLong test for comparison of each measure versus the LFC as reference

Thank you again for your thoughtful comments.

<Response for Reviewer #2>

1. It remains unclear what phenotype(s) of HF was /were included in the study.

Thank you for your important comment. Following your advice, we reviewed our data and updated our results. Among the 50 patients with HF, there were 15 patients with preserved EF, 15 with mid-range/mildly reduced EF, and 20 with reduced EF.

Additionally, we examined whether the PhA differences observed in the HF group were consistent across these HF phenotypes. Consistent with our original findings, all subgroups demonstrated lower PhA across all segments compared to controls. Especially, whole-body and lower limb PhA were significantly lower across all phenotypes.

However, as noted in your comment #3 and our response, we were cautious about interpreting the phenotype-specific differences in PhA more in detail due to the small sample size. Therefore, we have added these findings to the Supporting information (S1 Table, S2 Table), and briefly described in Results section as follows:

(Page 8/Lines 153-155) Among the HF group, 15 patients (30%) had preserved ejection fraction (EF), 15 patients (30%) had mid-range/mildly reduced EF, and 20 patients (40%) had reduced EF (S1 Table).

(Page 10/Lines 171-172) Additionally, differences in whole-body and lower limb PhA measured at 50 kHz were robust across all HF phenotypes (S2 Table).

2. How did the serum NT-proBNP concentration change in observation period and did the authors evaluate the link between the phase angle and NT-proBNP.

Yes, we evaluated the correlation between PhA and NT-proBNP, which showed a moderate correlation, particularly at 50 kHz. The results are presented in Figure 1 and supporting information S3 Table. However, as our study only measured markers at the time of admission, we were unable to collect longitudinal data to assess changes over time. We agree that tracking these changes could have provided a deeper understanding of PhA and its clinical implications. We have added this as a limitation in the study, as follows:

(Page 15/Lines 285-289) Third, the cross-sectional design of our study limited our ability to evaluate whether PhA or other markers, such as NT-proBNP, measured at a single time point or tracked over time, could offer insights into clinical outcomes and prognosis in HF patients. Further studies utilizing time-series data are needed to understand the clinical implications of PhA, both in the preclinical stage and over the course of long-term management.

3. The patients' cohort was heterogenic and composes of AF individuals and diabetes mellitus. Did the authors investigate whether the phase angle is able to predict HF exacerbation in individuals as with lower NT-proBNP (i.e., patients with diabetes) as well as individuals with higher than expected (AF patients)?

Thank you for raising this important issue. As you mentioned, external factors such as diabetes (DM) and atrial fibrillation (AF) can influence the expected changes in NT-proBNP levels in patients with acute HF due to their unique physiological character

---

## [Decision Letter · Decision Letter 1]

6 Dec 2024

PONE-D-24-38284R1Phase angle in bioelectrical impedance analysis for assessing congestion in acute heart failurePLOS ONE

Dear Dr. Lee,

Thank you for submitting your manuscript to PLOS ONE. After careful consideration, we feel that it has merit but does not fully meet PLOS ONE’s publication criteria as it currently stands. Therefore, we invite you to submit a revised version of the manuscript that addresses the points raised during the review process.

We look forward to receiving your revised manuscript.

Kind regards,

Rafael Oliveira

Academic Editor

PLOS ONE

Journal Requirements:

Additional Editor Comments:

Dear authors,

Thank you for improving your manuscript.

The reviewers were already satisfied with all answers and changes. However, I found minor details that were not properly addressed. Consequently, I would like to request your attention for the following points:

-please revise your writing and avoid doing it in the first person;

-a hypothesis should be added in the end of introduction (it should be supported with references);

-information about the procedures adopted before the body compositon analysis should be described as well as the timming of the day for the assessment;

-as suggested, authors already introduced a subsection in discussion with limitations title, but this title should also include practical implications and directions for future studies.

Best regards

Reviewers' comments:

Reviewer's Responses to Questions

**Comments to the Author**

1. If the authors have adequately addressed your comments raised in a previous round of review and you feel that this manuscript is now acceptable for publication, you may indicate that here to bypass the “Comments to the Author” section, enter your conflict of interest statement in the “Confidential to Editor” section, and submit your "Accept" recommendation.

Reviewer #2: All comments have been addressed

2. Is the manuscript technically sound, and do the data support the conclusions?

Reviewer #2: Yes

3. Has the statistical analysis been performed appropriately and rigorously? 

Reviewer #2: Yes

4. Have the authors made all data underlying the findings in their manuscript fully available?

Reviewer #2: Yes

5. Is the manuscript presented in an intelligible fashion and written in standard English?

Reviewer #2: Yes

6. Review Comments to the Author

Reviewer #2: The authors resubmitted a revised version of the paper along with thoroughly prepared respond to the reviwers. I am satisfied with the revision and have no serious concerns about the paper in its revised version.

7. PLOS authors have the option to publish the peer review history of their article (what does this mean?). If published, this will include your full peer review and any attached files.

Reviewer #2: No

---

## [Author Response · Author response to Decision Letter 1]

18 Dec 2024

We sincerely appreciate your considerate opinions. Our point-by-point responses to the comments with revised manuscript body in blue are described as follows:

Journal Requirements:

Please review your reference list to ensure that it is complete and correct. If you have cited papers that have been retracted, please include the rationale for doing so in the manuscript text, or remove these references and replace them with relevant current references.

→ We have carefully reviewed the reference list for completeness and retraction status. At this time, no retractions are noted, and the original reference list remains unchanged. Minor changes such as title change have been updated.

Additional Editor Comments:

- please revise your writing and avoid doing it in the first person;

→ Following the necessary changes in this round of minor revisions, we also went through language editing service for a secondary review to improve the flow and readability of the manuscript, as well as to ensure compliance with the request to avoid first-person tense. The certificate of language editing is attached in the revised file. We appreciate the opportunity to further enhance the quality of our submission.

- a hypothesis should be added in the end of introduction (it should be supported with references);

→ We have added the study hypothesis in the end of introduction as follows:

(Page 5/Lines 89-91) It was hypothesized that the phase angle is significantly correlated with NT-proBNP and LFC and may serve as a useful parameter for assessing congestion in patients with acute heart failure.

- information about the procedures adopted before the body compositon analysis should be described as well as the timming of the day for the assessment;

→ We regret not providing more detail on this point earlier. In fact, no additional procedures were adopted before the BIA testing, aside from resting on the testing bed, which has already been mentioned in the manuscript. To clarify this further, we have added some more details to the corresponding section, as follows:

(Page 7/Lines 118-119) BIA was performed within 3 days of admission, and no specific preparatory procedures were required.

- as suggested, authors already introduced a subsection in discussion with limitations title, but this title should also include practical implications and directions for future studies.

→ Thank you for your valuable comment and suggestion. As explained in our previous response letter, we opted to keep the clinical implications and directions for future studies within their respective contexts to maintain logical flow and readability throughout the discussion section (For example, “This suggests the potential utility of PhA as an assessment tool~” in lines 235-236, or “Further studies utilizing time-series data are required~” in lines 293-294).

However, to address your suggestion and ensure compliance as well, we have introduced the subtitle "Limitations" at the end of the discussion section, where the study limitations were originally structured separately.

We kindly ask for your understanding as we have put our best effort to balance these adjustments with the coauthors’ intentions.

Again, we deeply appreciate for your thoughtful comments.

---

## [Editor Report · Decision Letter 2]

22 Dec 2024

PONE-D-24-38284R2Phase angle in bioelectrical impedance analysis for assessing congestion in acute heart failurePLOS ONE

Dear Dr. Lee,

Thank you for submitting your manuscript to PLOS ONE. After careful consideration, we feel that it has merit but does not fully meet PLOS ONE’s publication criteria as it currently stands. Therefore, we invite you to submit a revised version of the manuscript that addresses the points raised during the review process.

We look forward to receiving your revised manuscript.

Kind regards,

Rafael Oliveira

Academic Editor

PLOS ONE

**Additional Editor Comments:**

Dear authors,

The majority of the comments were addressed with the exception of the following:

- information about the procedures adopted before the body compositon analysis should be described as well as the

timming of the day for the assessment;

→ We regret not providing more detail on this point earlier. In fact, no additional procedures were adopted before the

BIA testing, aside from resting on the testing bed, which has already been mentioned in the manuscript. To clarify this

further, we have added some more details to the corresponding section, as follows:

(Page 7/Lines 118-119) BIA was performed within 3 days of admission, and no specific preparatory procedures were

required.

I'm not sure about your answer. As the authors should know, the body composition analysis, using InBody S10, includes several cautions before measurement. The authors stated that no specific preparatory procedures were

required. However, such statement makes me wondering if the authors have enough expertise on using this device or measuring body compositon in general. So, I will provide another chance to clarify this (Cautions Before Measurement) as well as the timming of the day for assessment before moving with a final decision on your work. Authors must provide all information so other researchers can replicate your study in the exact same way.

Best regards

---

## [Author Response · Author response to Decision Letter 2]

24 Dec 2024

We sincerely appreciate your considerate opinions. Our point-by-point responses to the comments with revised manuscript body in blue are described as follows:

Additional Editor Comments:

The majority of the comments were addressed with the exception of the following:

- information about the procedures adopted before the body compositon analysis should be described as well as the timming of the day for the assessment;

→ We regret not providing more detail on this point earlier. In fact, no additional procedures were adopted before the BIA testing, aside from resting on the testing bed, which has already been mentioned in the manuscript. To clarify this further, we have added some more details to the corresponding section, as follows: 

(Page 7/Lines 118-119) BIA was performed within 3 days of admission, and no specific preparatory procedures were

required.

I'm not sure about your answer. As the authors should know, the body composition analysis, using InBody S10, includes several cautions before measurement. The authors stated that no specific preparatory procedures were required. However, such statement makes me wondering if the authors have enough expertise on using this device or measuring body compositon in general. So, I will provide another chance to clarify this (Cautions Before Measurement) as well as the timming of the day for assessment before moving with a final decision on your work. Authors must provide all information so other researchers can replicate your study in the exact same way.;

→ Thank you for your valuable feedback. We regret any inconvenience caused by not resolving this issue earlier. It seems there may have been a difference in understanding regarding the procedures adopted before the testing.

We initially understood your mention of "procedures" as referring to medical interventions (e.g., administering pre-operative antibiotics before surgery). For this reason, we previously replied that no preparatory medical procedures were required for BIA testing. And non-medical precautions necessary before the test were not included in our previous description of the BIA measurement steps, because we assumed these would be already familiar to physicians using the BIA in their practice.

However, we strongly agree that the manuscript should provide sufficient detail to allow any researchers to replicate our approach. Therefore, after carefully discussing your comment and reviewing other studies that incorporated BIA testing, we have thoroughly revised the BIA measurement section as follows:

(Page 7/Lines 118-126) Bioelectrical impedance analysis (BIA) was conducted within 3 days of admission to assess the edematous status of patients with acute HF. To reduce measurement bias, all tests were performed in a dedicated testing room by a single trained technician. Standardized pre-test precautions were implemented starting the day prior to the procedure. Patients were instructed to abstain from exercise, bathing, or activities leading to excessive sweating for 6–12 hours before the test and to avoid alcohol or caffeine for 24 hours prior. In the morning of the test, patients were asked to fast, empty their bladder, and remove all metallic or conductive items to prevent interference. BIA was performed with patients lying in the supine position on an insulated bed for 15 minutes to minimize measurement variability caused by fluid shifts.

We have put a great deal of thought and effort into understanding and addressing your comments as thoroughly as possible. However, we still have some concerns that our revisions may not fully reflect the intent of your feedback.

If you feel that our revisions have not adequately captured your intentions, we kindly ask for your understanding and would greatly appreciate it if you could provide more specific examples to guide us, allowing us another opportunity to make further improvements.

Again, we deeply appreciate for your thoughtful comments.

---

## [Editor Report · Decision Letter 3]

27 Dec 2024

Phase angle in bioelectrical impedance analysis for assessing congestion in acute heart failure

PONE-D-24-38284R3

Dear Dr. Sunki Lee,

We’re pleased to inform you that your manuscript has been judged scientifically suitable for publication and will be formally accepted for publication once it meets all outstanding technical requirements.

Kind regards,

Rafael Oliveira

Academic Editor

PLOS ONE

Additional Editor Comments (optional):

Dear authors,

Thank you for explaining in details all your toughts and also for dissipating my previous concerns. The present version of the manuscript was properly amended and my recommendation is to accept.

Congratulations!

Best regards
---

## [Editor Report · Acceptance letter]

15 Jan 2025

PONE-D-24-38284R3 

PLOS ONE

Dear Dr. Lee, 

I'm pleased to inform you that your manuscript has been deemed suitable for publication in PLOS ONE. Congratulations! Your manuscript is now being handed over to our production team.

Kind regards, 

on behalf of

Prof Rafael Oliveira 

Academic Editor

PLOS ONE